# GRAPH PARTITION NEURAL NETWORKS FOR SEMI-SUPERVISED CLASSIFICATION

## ABSTRACT

We present graph partition neural networks (GPNN), an extension of graph neural networks (GNNs) able to handle extremely large graphs. GPNNs alternate between locally propagating information between nodes in small subgraphs and globally propagating information between the subgraphs. To efficiently partition graphs, we experiment with spectral partitioning and also propose a modified multi-seed flood fill for fast processing of large scale graphs. We extensively test our model on a variety of semi-supervised node classification tasks. Experimental results indicate that GPNNs are either superior or comparable to state-of-the-art methods on a wide variety of datasets for graph-based semi-supervised classification. We also show that GPNNs can achieve similar performance as standard GNNs with fewer propagation steps.

## 1 INTRODUCTION

Graphs are a flexible way of encoding data, and many tasks can be cast as learning from graph-structured inputs. Examples include prediction of properties of chemical molecules (Duvenaud et al., 2015), answering questions about knowledge graphs (Marino et al., 2016), natural language processing with parse-structured inputs (trees or richer structures like Abstract Meaning Representations) (Banarescu et al.), predicting properties of data structures or source code in programming languages (Li et al., 2016), and making predictions from scene graphs (Teney et al., 2016). Sequence data can be seen as a special case of a simple chain-structured graph. Thus, we are interested in training high-capacity neural network-like models on these types of graph-structured inputs. Graph Neural Networks (GNNs) (Gori et al., 2005; Scarselli et al., 2009; Li et al., 2016; Qi et al., 2017; Li et al., 2017) are one of the best contenders, although there has been much recent interest in applying other neural network-like models to graph data, including generalizations of convolutional architectures (Duvenaud et al., 2015; Kipf & Welling, 2017). Gilmer et al. (2017) recently reviewed and unified many of these models.

An important issue that has not received much attention in GNN models is how information gets propagated across the graph. There are often scenarios in which information has to be propagated over long distances across a graph, e.g., when we have long sequences augmented with additional relationships between elements of the sequence, like in text, programming language source code, or temporal streams. The simplest approach, and the one adopted by almost all graph-based neural networks is to follow *synchronous message-passing systems* (Attiya & Welch, 2004) from distributed computing theory. Specifically, inference is executed as a sequence of rounds: in each round, every node sends messages to all of its neighbors, the messages are delivered and every node does some computation based on the received messages. While this approach has the benefit of being simple and easy to implement, it is especially inefficient when the task requires to spread information across long distances in the graph. For example, in processing sequence data, if we were to employ the above schedule for a sequence of length $N$, it would take $O(N^2)$ messages to propagate information from the beginning of the sequence to the end, and during training all $O(N^2)$ messages must be stored in memory. In contrast, the common practice with sequence data is to use a forward pass followed by a backward pass at a cost of $O(N)$ to propagate information from end to end, as in bidirectional recurrent neural networks (RNNs), for example.

One possible approach for tackling this problem is to propagate information over the graph following some pre-specified sequential order, as in Bidirectional LSTMs. However, this sequential

solution has several issues. First, if a graph used for training has large diameter, the unrolled GNN computational graph will be large (cf. Bidirectional LSTMs on long sequences). This leads to fundamental issues with learning (e.g., vanishing/exploding gradients) and implementation difficulties (i.e., resource constraints). Second, sequential schedules are typically less amenable to efficient acceleration on parallel hardware. More recently, Gilmer et al. (2017) attempted to tackle the first problem by introducing a "dummy node" with connections to all nodes in the input graph, meaning that all nodes are at most two steps away from each other. However, we note that the graph structure itself often contains important information, which is modified by adding additional nodes and edges.

In this work, we propose graph partition neural networks (GPNN) that exploit a propagation schedule combining features of synchronous and sequential propagation schedules. Concretely, we first partition the graph into disjunct subgraphs and a cut set, and then alternate steps of synchronous propagation within subgraphs with synchronous propagation within the cut set. In Sect. 2, we discuss different propagation schedules on an example, showing that GPNNs can be substantially more efficient than standard GNNs, and then present our model formally. Finally, we evaluate our model in Sect. 4 on a variety of semi-supervised classification benchmarks. The empirical results suggest that our models are either superior to or comparable with state-of-the-art learning systems on graphs.

## 2 MODEL

In this section, we briefly recapitulate graph neural networks (GNNs) and then describe our graph partition neural networks (GPNN). A graph $\mathcal{G} = (\mathcal{V}, \mathcal{E})$ has nodes $\mathcal{V}$ and edges $\mathcal{E} \subseteq \mathcal{V} \times \mathcal{V}$. We focus on directed graphs, as our approach readily applies to undirected graphs by splitting any undirected edge into two directed edges. We denote the out-going neighborhood as $\mathcal{N}_{out}(v) = \{u \in \mathcal{V} \mid (v, u) \in \mathcal{E}\}$, and similarly, the incoming neighborhood as $\mathcal{N}_{in}(v) = \{u \in \mathcal{V} \mid (u, v) \in \mathcal{E}\}$. We associate an edge type $c_{(v,u)} \in \{1, \dots, C\}$ with every edge $(v, u)$, where $C$ is some pre-specified total number of edge types. Such edge types are used to encode different relationships between nodes. Note that one can also associate multiple edge types with the same edge which results in a multi-graph. W.l.o.g. we assume one edge type per directed edge to simplify the notation.

### 2.1 GRAPH NEURAL NETWORKS

Graph neural networks (Scarselli et al., 2009; Li et al., 2016) can be viewed as an extension of recurrent neural networks (RNNs) to arbitrary graphs. Each node $v$ in the graph is associated with an initial state vector $\boldsymbol{h}_v^{(0)}$ at time step 0. Initial state vectors can be observed features or annotations as in Li et al. (2016). At time step $t$, an outgoing message is computed for each edge by transforming the source state according to the edge type, i.e.,

$$\boldsymbol{m}_{(v,u)}^{(t)} = M_{c_{(v,u)}}(\boldsymbol{h}_v^{(t)}), \tag{1}$$

where $M_{c_{(u,v)}}$ is a message function, which could be the identity or a fully connected neural network. Note the subscript $c_{(v,u)}$ indicating that different edges of the same type share the same instance of the message function. We then aggregate all messages at the receiving nodes, i.e.,

$$\bar{\boldsymbol{m}}_u^{(t)} = A(\{\boldsymbol{m}_{(v,u)}^{(t)} \mid v \in \mathcal{N}_{in}(u)\}), \tag{2}$$

where $A$ is the aggregation function, which may be a summation, average or max-pooling function. Finally, every node will update its state vector based on its current state vector and the aggregated message, i.e.,

$$\boldsymbol{h}_v^{(t+1)} = U(\boldsymbol{h}_v^{(t)}, \bar{\boldsymbol{m}}_v^{(t)}), \tag{3}$$

where $U$ is the update function, which may be a gated recurrent unit (GRU), a long short term memory (LSTM) unit, or a fully connected network. Note that all nodes share the same instance of update function. The described propagation step is repeatedly applied for a fixed number of time steps $T$, to obtain final state vectors $\{\boldsymbol{h}_v^{(T)} \mid v \in \mathcal{V}\}$. A node classification task can then be implemented by feeding these state vectors to a fully connected neural network which is shared by all nodes. Back-propagation through time (BPTT) is typically adopted for learning the model.

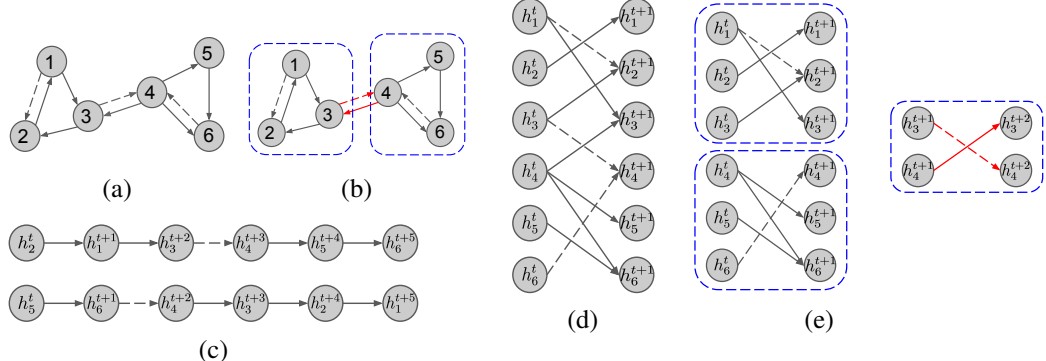

Figure 1: Propagation schedules on an example graph. (a) The input graph where the line type, i.e., solid & dash, indicates different edge types; (b) Graph partitions where blue bounding boxes indicate different subgraphs and red edges belong to the cut; (c) Computational graphs of two possible sequential propagation schedules of the input graph; (d) Computational graph for synchronous propagation schedule; (e) Computational graph for GPNNs where both inter-subgraph and intra-subgraph propagation steps are 1.

---

**Algorithm 1** Graph Partition Propagation Schedule.

1: **Input**: $K$ subgraphs $\{\mathcal{S}_k | k = 1, \ldots, K\}$, cut $\mathcal{S}_0$, outer propagation step limit $T$, intra-subgraph and inter-subgraph propagation step limits $T_S$ and $T_C$.
2: **for** $t = 1, \ldots, T$ **do**
3:     **for all** $k \in \{1, \ldots, K\}$ **do in parallel**
4:         Call SYNCPROP within subgraph $\mathcal{S}_k$ for $T_S$ steps.
5:     Call SYNCPROP within cut $\mathcal{S}_0$ for $T_C$ steps.
6: **function** SYNCPROP
7:     Compute & send messages as in Eq. (1)
8:     Aggregate messages as in Eq. (2)
9:     Update states as in Eq. (3)

---

## 2.2 GRAPH PARTITION NEURAL NETWORKS

The above inference process is described from the perspective of an individual node. If we look at the same process from the graph view, we observe a *synchronous schedule* in which all nodes receive and send messages at the same time, cf. the illustration in Fig. 1(d). A natural question is to consider different propagation schedules in which not all nodes in the graph send messages at the same time, e.g., *sequential schedules*, in which nodes are ordered in some linear sequence and messages are set only from one node at a time. A mix of the two ideas leads to our Graph Partition Neural Networks (GPNN), which we will discuss before elaborating on how to partition graphs appropriately. Finally, we discuss how to handle initial node labels and node classification tasks.

**Propagation Model**   We first consider the example graph in Fig. 1 (a). A corresponding computational graph that shows how information is propagated from time step $t$ to time step $t + 1$ using the standard (synchronous) propagation schedule is shown in Fig. 1 (d). The example graph's diameter is 5, and it hence requires at least 5 steps to propagate information over the graph. Fig. 1(c) instead shows two possible sequences that show how information can be propagated between nodes 2 to 6 and 5 to 1. These visualizations show that (i) a full synchronous propagation schedule requires significant computation at each step, and (ii) a sequential propagation schedule, in which we only propagate along sequences of nodes, results in very sparse and deep computational graphs. Moreover, experimentally, we found sequential schedules to require multiple propagation rounds across the whole graph, resulting in an even deeper computational graph.

In order to achieve both efficient propagation and tractable learning, we propose a new propagation schedule that follows a divide and conquer strategy. In particular, we first partition the graph into disjunct subgraphs. We will explain the details of how to compute graph partitions below. For now, we assume that we already have $K$ subgraphs such that each subgraph contains a subset of nodes

and the edges induced by this subset. We will also have a cut set, i.e., the set of edges that connect different subgraphs. One possible partition is visualized in Fig. 1 (b).

In GPNNs, we alternate between propagating information in parallel local to each subgraph (making use of highly parallel computing units such as GPUs) and propagating messages between subgraphs. Our propagation schedule is shown in Alg. 1. To understand the benefit of this schedule, consider a broadcasting problem over the example graph in Fig. 1. When information from any one node has reached all other nodes in the graph for the first time, this problem is considered as solved. We will compare the number of messages required to solve this problem for different propagation schedules.

*Synchronous propagation*: Fig. 1(d) shows that a synchronous step requires 10 messages. Broadcasting requires sufficient propagation steps to cover the graph diameter (in this case, 5), giving a total of $5 \times 10 = 50$ messages.

*Partitioned propagation*: For simplicity, we analyze the case $T_S = D_S, T_C = 1$, where $D_S$ is the maximum diameter of the subgraphs. Using the partitioning in 1(e), we have $D_S = 2$ and each step of intra-subgraph propagation requires 8 messages. After $T_S$ steps ($8D_S$ messages) the broadcast problem is solved within each subgraph. Inter-subgraph propagation requires 2 messages in this example, giving $8D_S + 2$ messages per outer loop iteration in Alg. 1. The example requires 2 outer iterations to broadcast between all subgraphs, giving a total of $2(8D_S + 2) = 36$ messages.

In general, our propagation schedule requires no more messages than the synchronous schedule to solve broadcast (if the number of subgraphs $K$ is set to 1 or $N$ then our schedule reduces to the synchronous one). We analyze the number of messages required to solve the broadcast problem on chain graphs in detail in Sect. A.1. Overall, our method avoids the large number of messages required by synchronous schedules, while avoiding the very deep computational graphs required by sequential schedules. Our experiments in Sect. 4 show that this makes learning tractable even on extremely large graphs.

**Graph Partition** We now investigate how to construct graph partitions. First, since partition problems in graph theory typically are NP-hard, we are only looking for approximations in practice. A simple approach is to re-use the classical spectral partition method. Specifically, we follow the normalized cut method in Shi & Malik (2000) and use the random walk normalized graph Laplacian matrix $L = I - D^{-1}W$, where $I$ is the identity matrix, $D$ is the degree matrix and $W$ is the weight matrix of graph (i.e., the adjacency matrix if no weights are presented).

However, the spectral partition method is slow and hard to scale with large graphs (Von Luxburg, 2007). For performance reasons, we developed the following heuristic method based on a multi-seed flood fill partition algorithm as listed in Alg. 2. We first randomly sample the initial seed nodes biased towards nodes which are labeled and have a large out-degree. We maintain a global dictionary assigning nodes to subgraphs, and initially assign each selected seed node to its own subgraph. We then grow the dictionary using flood fill, attaching unassigned nodes that are direct neighbors of a subgraph to that graph. To avoid bias towards the first subgraph, we randomly permute the order in the beginning of each round. This procedure is repeatedly applied until no subgraph grows anymore. There may still be disconnected components left in the graph, which we assign to the smallest subgraph found so far to balance subgraph sizes.

**Node Features & Classification** In practice, problems using graph-structured data sometimes (1) do not have observed features associated with every node (Grover & Leskovec, 2016); (2) have very high dimensional sparse features per node (Bing et al., 2015). We develop two types of models for the initial node labels: *embedding-input* and *feature-input*. For *embedding-input*, we introduce learnable node embeddings into the model to solve challenge (1), inspired by other graph embedding methods. For nodes with observed features we initialize the embeddings to these observations, and all other nodes are initialized randomly. All embeddings are fed to the propagation model and are treated as learnable parameters. For *feature-input*, we apply a sparse fully-connected network to input features to tackle challenge (2). The dimension-reduced feature is then fed to the propagation model, and the sparse network is jointly learned with the rest of model.

We also empirically found that concatenating the input features with the final embedding produced by the propagation model is helpful in boosting the performance.

---

**Algorithm 2** Modified Multi-seed Flood Fill Partition Algorithm.

---

1: **Input**: Graph $G$, number of subgraphs $K$, indices $I$ of nodes which are labeled.
2: Create two dictionaries $D$ and $L$ and $K$ FIFO queues $Q = \{Q_1, \ldots, Q_K\}$. $D$ maps node index to FALSE and $L$ maps node index to subgraph index 0.
3: $\forall u \in I$, compute the out-going degree $d_u$ of node $u$.
4: $\forall u \in I$, compute the probability $p_u = d_u / \sum_{v \in I} d_v$.
5: Sample $K$ nodes $S = \{s_1, \ldots, s_K\}$ from $I$ based on the above probability distribution $p$.
6: $\forall s_k \in S$, enqueue $s_k$ to $Q_k$, $D(s_k) = $ TRUE, $L(s_k) = k$.
7: **while** Any queue in $Q$ is not empty **do**
8:     **for** $k \in$ RANDPERM(K) **do**
9:         **if** $Q_k$ is not empty **then**
10:             $u \leftarrow$ pop $Q_k$
11:             **for** $v \in$ CHILDREN$(u)$ **do**
12:                 **if** $D(v) == $ FALSE **then**
13:                     Enqueue $v$ to $Q_k$
14:                     $L(v) = k$
15:                     $D(v) = $ TRUE
16: Put any unvisited nodes into the smallest subgraph and set $L$ accordingly.
17: Return $L$

---

## 3 RELATED WORK

There are many neural network models for handling graph-structured inputs. They can be roughly categorized into generalizations of recurrent neural networks (RNNs) (Goller & Kuchler, 1996; Gori et al., 2005; Scarselli et al., 2009; Socher et al., 2011b; Tai et al., 2015; Li et al., 2016; Marino et al., 2016; Qi et al., 2017; Li et al., 2017) and generalizations of convolutional neural networks (CNNs) (Bruna et al., 2014; Duvenaud et al., 2015; Kipf & Welling, 2017; Schlichtkrull et al., 2017). Gilmer et al. (2017) provide a good review and unification of many of these models, and they present some additional model variations that lead to strong empirical results in making predictions from chemical-structured inputs.

In RNN-like models, the standard approach is to propagate information using a synchronous schedule. In convolution-like models, the node updates mimic standard convolutions where all nodes in a layer are updated as functions of neighboring node states in the previous layer. This leads to information propagating across the graph in the same pattern as synchronous schedules. While our focus has been mainly on the RNN-like model of Li et al. (2016), it would be interesting to apply our schedules to the other models as well.

Some of the RNN based neural network models operate on restricted classes of graphs and employ sequential or sequential-like schedules. For example, recursive neural networks (Goller & Kuchler, 1996; Socher et al., 2011a) and tree-LSTMs Tai et al. (2015) have bidirectional variants that use fully sequential schedules.

It is possible to view Sukhbaatar et al. (2016) as a GNN model with a sequential schedule, where messages are passed inwards towards a master node that aggregates messages from different agents, and then outwards from the master node to all the agents. The difference in our work is the focus on graphs with arbitrary structure (not necessarily a sequence or tree). Recently, Marino et al. (2016) developed an attention-like mechanism to dynamically select a subset of graph nodes to propagate information from, but the propagation is synchronous amongst selected nodes.

An area where scheduling has been studied extensively is in the belief propagation (BP) literature. It is common to decompose a graph into spanning trees and sequentially update the tree structures Wainwright et al. (2002). See also Elidan et al. (2006); Tarlow et al. (2011); Sutton & McCallum (2012) for more discussion of sequential updates in the context of belief propagation. Finally, the question of sequential versus synchronous updates arises in numerical linear algebra. Jacobi iteration uses a synchronous update while Gauss-Seidel applies the same algorithm but according to a sequential schedule.

| Dataset | Type | #Nodes | #Edges | #Classes | #Features | Label Rate |
|---|---|---|---|---|---|---|
| Citeseer | Citation network | 3,327 | 4,732 | 6 | 3,703 | 0.036 |
| Cora | Citation network | 2,708 | 5,429 | 7 | 1,433 | 0.052 |
| Pubmed | Citation network | 19,717 | 44,338 | 3 | 500 | 0.003 |
| NELL | Knowledge graph | 65,755 | 266,144 | 210 | 5,414 | 0.1, 0.01, 0.001 |
| DIEL | Entity & list graph | 4,373,008 | 4,464,261 | 4 | 1,233,598 | 0.0095* |

Table 1: Dataset statistics. * indicates the average label rate over 10 fixed splits.

| Method | (Source) | Citeseer | Cora | Pubmed | NELL 10% | NELL 1% | NELL 0.1% |
|---|---|---|---|---|---|---|---|
| Feat | (Yang et al., 2016) | 57.2 | 57.4 | 69.8 | 62.1 | 40.4 | 21.7 |
| ManiReg | (Belkin et al., 2006) | 60.1 | 59.5 | 70.7 | 63.4 | 41.3 | 21.8 |
| SemiEmb | (Weston et al., 2012) | 59.6 | 59.0 | 71.1 | 65.4 | 43.8 | 26.7 |
| LP | (Zhu et al., 2003) | 45.3 | 68.0 | 63.0 | 71.4 | 44.8 | 26.5 |
| DeepWalk | (Perozzi et al., 2014) | 43.2 | 67.2 | 65.3 | 79.5 | 72.5 | 58.1 |
| ICA | (Lu & Getoor, 2003) | 69.1 | 75.1 | 73.9 | – | – | – |
| Planetoid (Transductive) | (Yang et al., 2016) | 64.9 | 75.7 | 75.7 | 84.5 | **75.7** | 61.9 |
| Planetoid (Inductive) | (Yang et al., 2016) | 64.7 | 61.2 | 77.2 | 70.2 | 59.8 | 45.4 |
| GCN | (Kipf & Welling, 2017) | **70.3** | 81.5 | 79.0 | †83.0 | †67.0 | †54.2 |
| GGNN* | (Li et al., 2016) | 68.1 | 77.9 | 77.2 | **84.6** | 66.2 | 59.1 |
| **GPNN** | (ours) | 69.7 | **81.9** | **79.2** | 83.7 | 74.6 | **63.1** |

Table 2: Classification accuracies on citation networks and knowledge graphs. * and † indicate we run our own (resp. the released) implementation..

## 4 EXPERIMENTS

We test our model on a variety of semi-supervised tasks: document classification on citation networks; entity classification in a bipartite graph extracted from a knowledge graph; and distantly-supervised entity extraction. We then compare different partition methods exploited by our model. We also compare the effectiveness of different propagation schedules. We follow the datasets and experimental setups in Yang et al. (2016). The statistics are summarized in Tab. 1, revealing that the datasets vary a lot in terms of scale, label rate and feature dimension. We report the details of hyper-parameters for all experiments in the appendix.

### 4.1 CITATION NETWORKS

We first discuss experimental results on three citation networks: Citeseer, Cora and Pubmed (Sen et al., 2008). The datasets contain sparse bag-of-words feature vectors for each document and a list of citation links between documents. Documents and citation links are regarded as nodes and edges while constructing the graph. 20 instances are sampled for each class as labeled data, 1000 instances as test data, and the rest are used as unlabeled data. The goal is to classify each document into one of the predefined classes. We use the same data split as in Yang et al. (2016) and Kipf & Welling (2017). We use an additional validation set of 500 labeled nodes for tuning hyperparameters as in Kipf & Welling (2017).

The results are listed in Thm. 2. We report the results of baselines directly from Yang et al. (2016) and Kipf & Welling (2017). We see that GPNN is on par with other state-of-the-art methods on these small graphs. We also conducted experiments with 10 random splits and results are reported in the appendix. We found these datasets easy to overfit due to their small size, and use *feat-input* rather than *embedding-input*, as the latter case increases the model capacity as well as the risk of overfitting. We also show a t-SNE (Maaten & Hinton, 2008) visualization of node representations produced by the propagation model of GGNN and GPNN on the Cora dataset in Fig. 2 (a) and (b) respectively. The visualizations show that the node representations of GPNN are better separated.

| Method | Recall@k |
|---|---|
| LP   (Zhu et al., 2003) | 16.20 |
| DeepWalk   (Perozzi et al., 2014) | 25.80 |
| Feat   (Yang et al., 2016) | 34.90 |
| DIEL   (Bing et al., 2015) | 40.50 |
| ManiReg   (Belkin et al., 2006) | 47.70 |
| SemiEmb   (Weston et al., 2012) | 48.60 |
| Planetoid (Transductive)   (Yang et al., 2016) | 50.00 |
| Planetoid (Inductive)   (Yang et al., 2016) | 50.10 |
| GGNN*   (Li et al., 2016) | 51.15 |
| GPNN | **52.11** |

Table 3: Average recall on the DIEL dataset. Note that GCN is not included as it runs out of memory.

## 4.2   ENTITY CLASSIFICATION

Next, we consider experimental results of entity classification task on the NELL dataset extracted from the knowledge graph first presented in Carlson et al. (2010). A knowledge graph consists of a set of entities and a set of directed edges which have labels (i.e., different types of relation). Following Yang et al. (2016), each triplet $(e_1, r, e_2)$ of entities $e_1, e_2$ and relation $r$ in the knowledge graph is split into two tuples. Specifically, we assign separate relation nodes $r_1$ and $r_2$ to each entity and thus obtain $(e_1, r_1)$ and $(e_2, r_2)$. Entity nodes are associated with sparse feature vectors. We follow Kipf & Welling (2017) to extend the number of features by assigning a unique one-hot representation for every relation node. This results in a 61278-dim sparse feature vector per node. An additional validation set of 500 labeled nodes under the label rate 0.1% as in Kipf & Welling (2017) is used for tuning hyperparameters. The chosen hyperparameters are then used for other label rates. The semi-supervised task here considers three different label rates 10%, 1%, 0.1% per class in the training set. We run the released code of GCN with the reported hyperparameters in Kipf & Welling (2017). Since we did not observe overfitting on this dataset, we choose the *embedding-input* variant as the input model. The results are shown in Tab. 2, where we see that our model outperforms competitors under the most challenging label rate 0.001 and obtain comparable results with the state of the art on other label rates.

## 4.3   DISTANTLY-SUPERVISED ENTITY EXTRACTION

Finally, we consider the DIEL (Distant Information Extraction using coordinate-term Lists) dataset (Bing et al., 2015). This dataset constructs a bipartite graph where nodes are medical entities and texts (referred as mentions and coordinate lists in the original paper). Texts contain some facts about the medical entities. Edges of the graph are links between entities and texts. Each entity is associated with a pre-extracted sparse feature vector. The goal is to extract medical entities from text given sparse feature vectors and the graph. As shown in Tab. 1, this dataset is very challenging due to its extremely large scale and very high-dimensional sparse features. Note that we attempted to run the released code GCN model on this dataset, but ran out of memory. We follow the exact experimental setup as in Bing et al. (2015); Yang et al. (2016), including 10 different data splits, preprocessing of entity mentions and coordinate lists, and evaluation. We randomly sample $1/5$ of the training nodes as the validation set. We regard the top-$k$ entities returned by a model as positive instances and compute recall@$k$ as the evaluation metric where $k = 240000$ as in Bing et al. (2015); Yang et al. (2016). Average recall over 10 runs is reported in Tab. 3, and we see that GPNN outperforms all other models. Note that since Freebase is used as ground truth and some entities are not present in texts, the upper bound of recall given by Bing et al. (2015) is 0.617.

## 4.4   COMPARISON OF DIFFERENT PARTITION METHODS

We now compare the two partition methods we considered for our model: spectral partition and our modified multi-seed flood fill. We use the NELL data set to benchmark and report the average validation accuracy over 10 runs in Tab. 4, in which we also report the average runtime of the partition process. The accuracies of the trained models do not allow for a clear conclusion as to

| Method | 5 | 10 | 20 | 30 |
|---|---|---|---|---|
| Spectral Partition | 54.8 (2.49s) | 55.6 (4.16s) | 58.0 (12.2s) | 60.1 (3115s) |
| Modified Multi-seed Flood Fill | 62.0 (0.36s) | 63.1 (0.36s) | 57.5 (0.43s) | 59.9 (0.23s) |

Table 4: Accuracy and run time of different partition methods with different numbers of subgraphs.

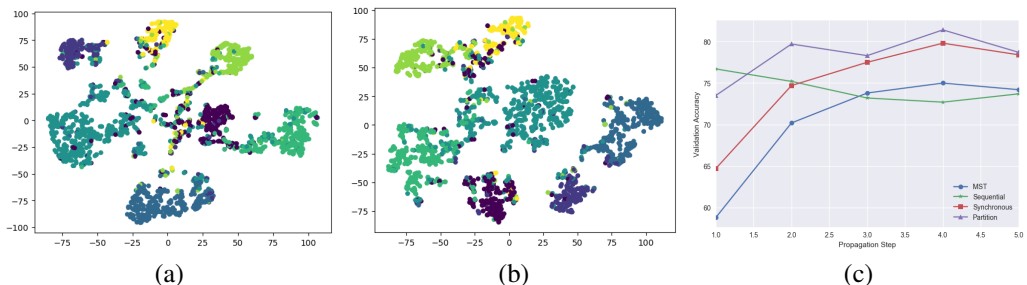

Figure 2: (a), (b) The t-SNE visualization of node representations produced by propagation model of GGNN and GPNN on Cora dataset in which nodes actually belong to 7 classes. (c) Comparison of different propagation schedules with varying propagation steps.

which method to use, and in our further experiments they seem to highly depend on the number of subgraphs, the connectivity of input graphs, optimization and other factors. However, our multi-seed flood fill partition method is substantially faster and is efficiently applicable to very large graphs.

### 4.5    COMPARISON OF DIFFERENT PROPAGATION SCHEDULES

Besides the synchronous and our partition based propagation schedules, we also investigated two further schedules based on a sequential order and a series of minimum spanning trees (MST).

To generate a sequential schedule, we first perform graph traversal via breadth first search (BFS) which gives us a visiting order. We then split the edges into those that follow the visiting order and those that violate it. The edges in each class construct a directed acyclic graph (DAG), and we construct a propagation schedule from each DAG following the principle that every node will send messages once it receives all messages from its parents and updates its own state. An example of the schedule is given in the appendix. Note that this sequential schedule reduces to a standard bidirectional recurrent neural network on a chain graph.

For the MST schedule, we find a sequence of minimum spanning trees as follows. We first assign random positive weights between 0 and 1 to every edge and then apply Kruskal's algorithm to find an MST. Next we increase the weights by 1 for edges which are present in the MST we found so far. This process is iterated until we find $K$ MSTs where $K$ is the total number of propagation steps.

We compare all four schedules by varying the number of propagation steps on the Cora dataset. The validation accuracies are shown in Fig. 2 (c). In these results, the meaning of one propagation step varies, so the takeaways are based just on the trends and overall performance across number of propagation steps. For the synchronous schedule, it means that every node sent and received messages once and updated its state. For the sequential schedule, it means that messages from all roots of the two DAGs were sent to all the leaves. For the MST-based schedule, it means sending messages from the root to all leaves on one minimum spanning tree. For our partition schedules, it means one outer loop of the algorithm. In this sense, messages are propagated furthest through the graph for the sequential schedule within one propagate step. This is also validated by the best performance of sequential schedule in the beginning. However, when increasing the number of propagation steps, it performs worse as the deep computational graph makes the learning problem very hard. Our partition schedule is better than other schedules when the number of propagation steps is small and tends to perform similarly with synchronous schedule with more steps.

## 5 CONCLUSION

We presented graph partition neural networks, which extend graph neural networks. Relying on graph partitions, our model alternates between locally propagating information between nodes in small subgraphs and globally propagating information between the subgraphs. Moreover, we propose a modified multi-seed flood fill for fast partitioning of large scale graphs. Empirical results show that our model performs better or is comparable to state-of-the-art methods on a wide variety of semi-supervised node classification tasks.

There are quite a few exciting directions to explore in the future. One is to learn the graph partitioning as well as the GNN weights, using a soft partition assignment. Other types of propagation schedules which have proven useful in probabilistic graphical models are also worthwhile to explore in the context of GNNs. To further improve the efficiency of propagating information, different nodes within the graph could share some memory, which mimics the shared memory model in the theory of distributed computing. Perhaps most importantly, this work makes it possible to run GNN models on very large graphs, which potentially opens the door to many new applications.

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

| Method | Citeseer | Cora | Pubmed |
|---|---|---|---|
| GCN[†]   (Kipf & Welling, 2017) | **68.7** $\pm$ 2.0 | **80.4** $\pm$ 2.8 | **77.5** $\pm$ 2.1 |
| GGNN[*]   (Li et al., 2016) | 66.3 $\pm$ 2.0 | 78.9 $\pm$ 2.6 | 74.7 $\pm$ 2.8 |
| GPNN | 68.6 $\pm$ 1.7 | 79.9 $\pm$ 2.4 | 76.1 $\pm$ 2.0 |

Table 5: Classification accuracies on citation networks with 10 random splits. [*] and [†] indicates we run our own implementation and the released code respectively.

# A   APPENDIX

## A.1   BI-DIRECTIONAL CHAIN

In this section, we revisit the broadcast problem on bi-direction chain graphs. We show that our propagation schedule has advantages over the synchronous one via the following proposition.

**Proposition 1.** *Let $\mathcal{G}$ be a bi-direction chain of size $N$. We have: (1) Synchronous propagation schedule requires $2(N-1)^2$ messages to solve the problem; (2) If we partition the chain evenly into $K$ sub-chains for $1 \leq K \leq N$, GPNN propagation schedule can solve the problem with $2((N-K)^2 + (K-1)^2)$ messages.*

*Proof.* We first analyze the case for synchronous propagation schedule. At each round, it needs $2(N-1)$ messages to propagate messages one step away. Since it requires at least $(N-1)$ steps for message from one endpoint of the chain to reach the other, the number of messages to solve broadcast is thus $2(N-1)^2$.

We now turn to our schedule. Since the chain is evenly partitioned, each sub-chain is of $n = N/K$ nodes. We need to perform $(n-1)$ propagation steps to traverse a sub-chain, so we set $T_S = n-1$. The number of messages required by a single sub-chain during the intra-subgraph propagation phase is $2(n-1)^2$, and so all $K$ sub-chains collectively require $2K(n-1)^2$ messages. Between intra-subgraph propagation, we perform $T_C = 1$ step of inter-subgraph propagation to transfer messages over the cut edges between sub-chains. Each inter-subgraph step requires 2 messages per cut edge - i.e. 2(K-1) messages in total. We need $K$ outer loops to ensure that message from any node can reach any other nodes, and strictly speaking, the the last inter-subgraph propagation step is unnecessary. So in total, we require $K \times 2K(n-1) + (K-1) \times 2(K-1) = 2((N-K)^2 + (K-1)^2)$ messages, which proves the proposition. $\square$

One can see from the above proposition that if we take $K = 1$ and $K = N$, the number of messages of our schedule matches the synchronous one. We can also derive the optimal value of $K$ as $(N + 1)/2$ resulting in a factor of 2 reduction in the total messages sent compared to the synchronous schedule.

## A.2   HYPERPARAMETERS

We train all models using Adam Kingma & Ba (2014) with a learning rate of 0.01. We also use early stopping with a window size of 10. We clip the norm gradient to ensure that it is no larger than 5.0. The maximum epochs for citation networks, NELL and DIEL are set to 200, 300 and 100 respectively. The weight decays for citation networks, NELL and DIEL are set to $5.0e^{-4}$, $1.0e^{-5}$ and $1.0e^{-3}$ respectively. The dimensions of state vectors of GPNNfor Cora, Citeseer, Pubmed, NELL and DIEL are set to 128, 128, 64, 512 and 64. The output model for Cora, Citeseer, NELL is just softmax layer. For Pubmed and DIEL, we add one hidden layer with $tanh$ activation function before the softmax which have dimension 512 and 2048 respectively.

## A.3   RANDOM SPLITS OF CITATION NETWORKS

We include the results on citation networks with 10 random splits in Table 5. From the table, we can see that our results are comparable with the state-of-the-art on these small scale datasets.

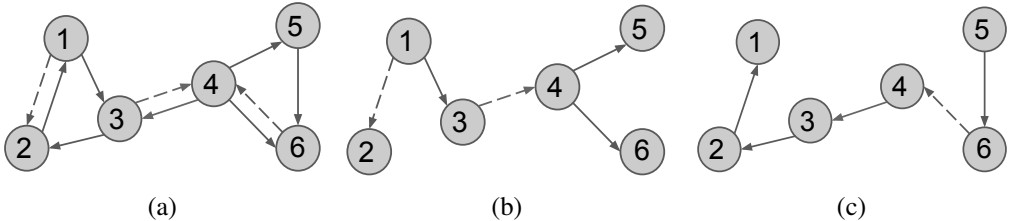

(a)          (b)          (c)

Figure 3: Sequential scheduling. (a) The original graph. (b) and (c) are the two DAGs obtained by the sequential schedule we described in section 4.5 where BFS traversal is started from node 1.

## A.4 SEQUENTIAL PROPAGATION SCHEDULE

In Fig. 3 we show an example visualization of the DAGs decomposition of the sequential propagation schedule we implemented in the section 4.5.

## A.5 IMPLEMENTATION

The released code of GGNN (Li et al., 2016) is implemented in Torch. We implement both our own version of GGNN and our model in Tensorflow (Abadi et al., 2015). To ensure correctness, we first reproduced the experimental results of the paper on bAbI artificial intelligence (AI) tasks with our implementations of GGNN. Our code will be released soon. One challenging part is the implementation of synchronous propagation within subgraphs. We implicitly implement the parallel part by building one separate branch of the computational graph for each subgraphs (i.e., use a Python `for` loop rather than `tf.while_loop`). This relies on the claim that tensorflow optimizes the execution of the computational graph in a way that independent branches of the graph will be executed in parallel as decribed in Abadi et al. (2015). However, since we have no control of the optimization of the computational graph, this part could be improved by explicitly putting each branch on one separate computation device, just like the multi-tower solution for training convolutional neural networks (CNNs) on multiple GPUs.

