# OpenReview forum: "Graph Partition Neural Networks for Semi-Supervised Classification"
_ICLR.cc/2018/Conference — Invite to Workshop Track_

### Official Review · AnonReviewer1 · 2017-11-26
**Extends the GNN framework to handle large graphs by running async updates on subgraphs derived by using graph partitioning algorithms. Results demonstrated on semi-supervised task.**

**Rating:** 6
**Confidence:** 3

**Review:**

Graph Neural Networks are methods using NNs to deal with graph data (each data point has some features, and there is some known connectivity structure among nodes) for problems such as semi-supervised classification. They can also be viewed as an abstraction and generalizations of RNNs to arbitrary graphs. As such they assume each unit has inputs from other nodes, as well as from some stored representation of a state and upon receiving all its information and executing a computation on the values of these inputs and its internal state, it can update the state as well as propagate information to neighbouring nodes.

This paper deals with the question of computing over very large input graphs where learning becomes computationally problematic (eg hard to use GPUs, optimization gets difficult due to gradient issues, etc). The proposed solution is to partition the graph into sub graphs, and use a schedule alternating between performing intra and inter graph partitions operations. To achieve that two things need to be determined - how to partition the graph, and which schedules to choose. The authors experiment with existing and somewhat modified solutions for each of these problems and present results that show that for large graphs, these methods are indeed effective and achieve state-of-the-art/improved results over existing methos.

The main critique is that this feels more of an engineering solution to running such GNNs on large graphs than a research innovations. The proposed algorithms are straight forward and/or utilize existing algorithms, and introduce many hyper parameters and ad-hoc decisions (the scheduling to choose for instance). In addition, they do not satisfy any theoretical framework, or proposed in the context of a theoretical framework than has guarantees of mathematical properties that are desirable. As such it is likely of use for practitioners but not a major research contribution.

---

> ### Author Response · Authors · 2017-12-31
> **Response:**
>
> We thank the reviewer for the valuable comments. Given the increasing popularity of graph neural networks, e.g., see recent references in A1 of Anonymous Reviewer 4, we believe it is still valuable to share our studies of graph partitioning and message-passing schedules with the ICLR community. As our results show, these approaches will be very important as the community starts considering larger graphs than those currently being investigated in the graph network literature.

---

### Official Review · AnonReviewer2 · 2017-11-27
**Partitioning for better message passing - maybe?**

**Rating:** 5
**Confidence:** 3

**Review:**

The authors investigate different message passing schedules for GNN learning.  Their proposed approach is to partition the graph into disjoint subregions, pass many messages on the sub regions and pass fewer messages between regions (an approach that is already considered in related literature, e.g., the BP literature), with the goal of minimizing the number of messages that need to be passed to convey information between all pairs of nodes in the network.  Experimentally, the proposed approach seems to perform comparably to existing methods (or slightly worse on average in some settings).  The paper is well-written and easy to read.  My primary concern is with novelty.  Many similar ideas have been floating around in a variety of different message-passing communities.  With no theoretical reason to prefer the proposed approach, it seems like it may be of limited interest to the community if speed is its only benefit (see detailed comments below).

Specific comments:

1)  "When information from any one node has reached all other nodes in the graph for the first time, this problem is considered as solved."

Perhaps it is my misunderstanding of the way in which GNNs work, but isn't the objective actually to reach a set of fixed point equations.  If so, then simply propagating information from one side of the graph may not be sufficient.

2)  The experimental results in Section 4.4 are almost impossible to interpret.  Perhaps it is better to plot number of edges updated versus accuracy?  This at least would put them on equal footing.   In addition, the experiments that use randomness should be repeated and plotted on average (just in case you happened to pick a bad schedule).

3)  More generally, why not consider random schedules (i.e., just pick a random edge, update, repeat) or random partitions?  I'm not certain that a fixed set will perform best independent of the types of updates being considered, and random schedules, like the fully synchronous case for an important baseline (especially if update speed is all you care about).

Typos:

-pg. 6, "Thm. 2" -> "Table 2"

---

> ### Author Response · Authors · 2017-12-29
> **Response:**
>
> We thank the reviewer for bringing up random schedules. We added the experiment as per suggestion.
>
> Q1: Reach a set of fixed point equations.
> A1: The original GNN paper (Scarselli et al. 2009) indeed requires that the state update function is a contraction map (and by Banach’s theorem thus has a fixed point). However, recent gated GNN adaptations (e.g. Li et al. 2015) drop this requirement and instead just fix a number of propagation steps as a hyperparameter; training and testing is then very similar to the RNN setting. We also follow the latter setting since (1) for a general nonlinear dynamic system, no guarantee can be made regarding whether fixed points can be reached; and (2) the learning algorithm, i.e., back-propagation through time (BPTT) would be significantly more time consuming as fixed-point convergence typically requires very many propagation steps, which is impractical for very large graphs. In the paper, we use a synthetic broadcasting problem to study the difference in efficiency of various message passing schedules in an idealized setting. As you observe, propagation across the whole graph may often not suffice to solve all tasks, but is a simple way to study if long-range dependencies between different vertices can be modeled at all.
>
> Q2: Experimental results in section 4.4.
> A2: We assume the reviewer has a typo here in an sense that you actually refer to section 4.5.
> Thanks for your suggestion of plotting number of edges updated versus accuracy. We will replot in the final version. To clarify, in Fig. 2 (c), assuming graph G(V, E) is singly connected, then the “# edges per propagation step” of MST, Sequential, Synchronous and Partition are |V|-1, |E|, |E| and |E|. We also attach the average results of 10 runs with different random seeds on Cora as below.
> -----------------------------------------------------------------------------------
> | Prop Step      | 1                      | 3                      | 5                     |
> -----------------------------------------------------------------------------------
> | MST                | 59.94 +- 0.89  | 71.83 +- 0.96 | 77.1 +- 0.72   |
> -----------------------------------------------------------------------------------
> | Sequential     | 73.04 +- 1.93  | 77.55 +- 0.65 | 74.89 +- 1.26 |
> -----------------------------------------------------------------------------------
> | Synchronous | 67.36 +- 1.44 | 80.15 +- 0.80 | 80.06 +- 0.98  |
> -----------------------------------------------------------------------------------
> | Partition         | 68.1 +- 1.98   | 80.27 +- 0.78 | 80.12 +- 0.93  |
> -----------------------------------------------------------------------------------
> We will plot the mean curve with error bar and improve the writing in the final version.
>
> Q3: Random and Synchronous Schedules
> A3: To clarify, we did compare with a fully synchronous schedule which is the one adopted by the GGNN model. Also, speed is not the only benefit, as with partition based schedules, memory is saved which enables us to apply the model to large-scale graph problems.
>
> Developing schedules that depend on the type of updates is a very interesting and promising direction. We will explore it in the future. On the other side, our schedule is not fixed in a sense that the partition depends the structure of input graph.
>
> We did an experiment on random schedules. In particular, for k-step propagation, we randomly sample 1/k proportion of edges from the whole edge set without replacement and use them for propagation. We summarize the results (10 runs) on the Cora dataset in the table below,
> --------------------------------------------------------
> | K            | 2         | 3        | 5         | 10      |
> --------------------------------------------------------
> | Avg Acc | 76.03 | 74.71 | 72.09 | 69.99 |
> --------------------------------------------------------
> | Std Acc  | 1.55   | 1.31   | 1.81   | 2.26   |
> --------------------------------------------------------
> From the results, we can see that the best average accuracy (K = 2) is 76.03 which is still lower than both synchronous and our partition based schedule. Note that this result roughly matches the one with spanning trees. The reason might be that random schedules typically need more propagation steps to spread information throughout the graph. However, more propagation steps of GNNs may lead to issues in learning with BPTT. Additional results on other datasets will be included in the final version.

---

### Official Review · AnonReviewer4 · 2017-12-02
**Incremental improvement on graph neural networks with heuristic graph partitioning**

**Rating:** 6
**Confidence:** 3

**Review:**

Since existing GNNs are not computational efficient when dealing with large graphs, the key engineering contributions of the proposed method, GPNN, are a partitioning and the associated scheduling components.

The paper is well written and easy to follow. However, related literature for message passing part is inadequate.

I have two concerns. The primary one is that the method is incremental and rather heuristic. For example, in Section 2.2, Graph Partition part, the authors propose to "first randomly sample the initial seed nodes biased towards nodes which are labeled and have a large out-degree", they do not give any reasons for the preference of that kind of nodes.

The second one is that of the experimental evaluation. GPNN is on par with other methods on small graphs such as citation networks, performs comparably to other methods, and only clearly outperforms on distantly-supervised entity extraction dataset. Thus, it is  not clear if GPNN is more effective than others in general. As for experiments on DIEL dataset, the authors didn't compare to GCN due to the simple reason that GCN ran out of memory. However, vanilla GCN could be trivially partitioned and propagating just as shown in this paper. I think such experiment is crucial, without which I cannot assess this method properly.

---

> ### Author Response · Authors · 2017-12-31
> **Response:**
>
> We thank the reviewer for the valuable comments. We did large-scale experiments with GCN as suggested.
>
> Q1: The method is incremental.
> A1: We agree that our contribution is an extension of earlier work. However, given the rapidly increasing interest in graph neural networks and their variants (cf. some recent references below), we believe studying methods to make them computationally effective is very valuable for the community. As GNNs operate on graphs that are often very different from common probabilistic graphical models (PGMs), the impact of different schedules in the two areas may be very different. For example, spanning tree based schedules are known to be very effective for PGMs. However, many graphs require a very large number of spanning trees to achieve satisfactory performance, which in turn seems to cause optimization problems (cf. the experimental results with minimal spanning trees in Sect. 4.5).
>
> Li, Y., Tarlow, D., Brockschmidt, M. and Zemel, R., 2016. Gated graph sequence neural networks. ICLR.
>
> Qi, X., Liao, R., Jia, J., Fidler, S. and Urtasun, R., 2017. 3d graph neural networks for rgbd semantic segmentation. ICCV.
>
> Li, R., Tapaswi, M., Liao, R., Jia, J., Urtasun, R. and Fidler, S., 2017. Situation Recognition with Graph Neural Networks. ICCV.
>
> Garcia, V. and Bruna, J., 2017. Few-Shot Learning with Graph Neural Networks. arXiv preprint arXiv:1711.04043.
>
> Bruna, J. and Li, X., 2017. Community Detection with Graph Neural Networks. arXiv preprint arXiv:1705.08415.
>
> Nowak, A., Villar, S., Bandeira, A. and Bruna, J. A Note on Learning Algorithms for Quadratic Assignment with Graph Neural Networks. arXiv preprint arXiv:1706.07450.
>
> Q2: Add literature of message passing.
> A2: We will add relevant work in PGMs. We plan to discuss the two papers below in an updated submission, but would be happy to incorporate more papers.
>
> Sontag, D. and Jaakkola, T., 2009, April. Tree block coordinate descent for MAP in graphical models. AISTATS.
>
> Komodakis, N., Paragios, N. and Tziritas, G., 2011. MRF energy minimization and beyond via dual decomposition. IEEE PAMI.
>
> Q3: Preference of nodes with high-degree.
> A3: We prefer the high-degree nodes as the seeds because in other graphs tasks (e.g., influence maximization in social networks), high-degree heuristics are shown to be a simple and yet strong baseline (cf. paper below). We will update the paper to make this reasoning clearer.
>
> Kempe, D., Kleinberg, J. and Tardos, É., 2003, August. Maximizing the spread of influence through a social network. In ACM SIGKDD.
>
> Q4: GCN on DIEL.
> A4: We ran a set of experiments of GCN on the DIEL dataset. This required a significant amount of engineering effort in the implementation. First, we use sparse operations in many places and reduce the feature dimension by introducing a learnable linear layer such that the model to fit into 128GB CPU memory. We then implemented a partition based schedule for GCN. In particular, we first get the partition using the proposed multi-seed flood fill method. Then we construct two graph laplacian matrices for the disconnected clusters and the cut, denoting as L_cluster and L_cut. In original GCN, a layer is expressed as ReLU( L * X * W ) where L, X and W are graph laplacian, node states and weight parameters respectively. In the partition based GCN, it is ReLU( L_cut * L_cluster * X * W ). We tuned hyperparameters and the results are summarized as below.
> ---------------------------------------------------------------------------
> Method       | GCN  | GCN + Partition | GGNN | GPNN |
> ---------------------------------------------------------------------------
> Avg. Recall  | 48.14 | 48.47                   | 51.15   | 52.11  |
> ---------------------------------------------------------------------------
>
> We observe that (1) both GCN and its partition variant are worse than that of GGNN and our GPNN; (2) partition based GCN has a marginal improvement over the vanilla one.
>
> One reason why GCN performs poorly is that it requires more layers to reach similar performance of GNN since a k-layer GCN will propagate messages k-hops away whereas GNN has the advantage of propagating more even within one layer. Directly adding more layers is infeasible here as it can not fit into memory. We tried to reduce the feature dimension in order to add more layers which leads to a new issue that the features may not be discriminative enough. We hypothesize that if we could add more layers to GCN without reducing the feature dimension too much, GCN will perform similarly. However, it requires more memory and/or intensive optimization of the code which we left as future work.
>
> The marginal gain of partition based GCN is understandable as the model just splits one linear transform L into L_cut and L_cluster without enhancing the model capacity significantly. Note that the sparsities of L * X and L_cut * L_cluster * X are different.
>
> Finally, our code based on Tensorflow will be released soon.

---

### Decision · Program_Chairs · 2018-01-29
**ICLR 2018 Conference Acceptance Decision**

**Decision:**

Invite to Workshop Track

**Comment:**

This paper was perceived as being well written, but the technical contribution was seen as being incremental and somewhat heuristic in nature. Some important prior work was not discussed and more extensive experimentation was recommended.
However, the proposed approach of partitioning the graph into sub graphs and a schedule alternating between intra and inter graph partitions operations has some merit.

The AC recommends inviting this paper to the Workshop Track.